# SUPERTWEETEVAL: A Challenging, Unified and Heterogeneous Benchmark for Social Media NLP Research

**Dimosthenis Antypas**[1]    **Asahi Ushio**[1]    **Francesco Barbieri**[2]    **Leonardo Neves**[2]
**Kiamehr Rezaee**[1]    **Luis Espinosa-Anke**[1,4]    **Jiaxin Pei**[3]    **Jose Camacho-Collados**[1]

[1] Cardiff NLP, Cardiff University, UK [2] Snap Inc., Santa Monica, CA, USA

[3] School of Information, University of Michigan, USA [4] AMPLYFI, UK

[1] {antypasd,ushioa,rezaeek,espinosa-ankel,camachocolladosj}@cardiff.ac.uk
[2] {fbarbieri,lneves}@snap.com    [3] pedropei@umich.edu

## Abstract

Despite its relevance, the maturity of NLP for social media pales in comparison with general-purpose models, metrics and benchmarks. This fragmented landscape makes it hard for the community to know, for instance, given a task, which is the best performing model and how it compares with others. To alleviate this issue, we introduce a unified benchmark for NLP evaluation in social media, SUPERTWEETEVAL, which includes a heterogeneous set of tasks and datasets combined, adapted and constructed from scratch. We benchmarked the performance of a wide range of models on SUPERTWEETEVAL and our results suggest that, despite the recent advances in language modelling, social media remains challenging.

## 1 Introduction

There is overwhelming evidence that general-purpose NLP systems suffer from a significant drop in performance when exposed to tasks in specialised domains. This has been shown in disparate domains such as the legal, medical and financial, to name a few. Specifically, for these domains, specialised language models (LMs) such as Legal-BERT, SciBERT, PubMedBERT, or BloombergGPT (Chalkidis et al., 2020; Beltagy et al., 2019; Gu et al., 2021; Wu et al., 2023) have shown to exhibit lower perplexity and higher downstream performance across the board. The need for specialised LMs and corresponding evaluation benchmarks is exacerbated in social media, where instead of (or rather, in addition to) a specialised domain, an NLP system has to deal with idiosyncrasies such as emoji (Miller et al., 2017; Cappallo et al., 2018; Barbieri et al., 2017a), poor capitalisation (Derczynski et al., 2015), vulgarisms and colloquialisms (Camacho-Collados et al., 2020), fast language change (Del Tredici et al., 2019; Loureiro et al., 2022a), and dynamic and platform-specific communicative structures (Bastos et al., 2013).

Therefore, unsurprisingly, a strand of Twitter-specific NLP research has produced what we would consider now *de-facto* models and datasets. On one hand, specialized LMs, either pre-trained on multilingual Twitter text alone (Nguyen et al., 2020; DeLucia et al., 2022; Barbieri et al., 2022b), or including social engagements (Zhang et al., 2022b); and on the other, joint Twitter-specific models *and datasets* such as TweetEval (Barbieri et al., 2020). However, one area where social media NLP research seems to be lacking behind is in matching with appropriate resources the current shift towards in-context learning (ICL) that Large LMs (LLMs) enable (Min et al., 2022). Benchmarks such as TweetEval, while helpful, are constrained to tweet classification tasks, crucially neglecting sequence tagging, generation and question answering (QA), for instance, which not only are more diverse, but better test beds for LLMs and comparing fine-tuning vs ICL (Liu et al., 2022).

The contributions of this paper can be summarized as follows. First, we introduce the SUPER-TWEETEVAL benchmark[1]. SUPERTWEETEVAL fills an important gap in the current NLP landscape by unifying diverse social media tasks and datasets beyond tweet classification (e.g., NER, question answering or tweet similarity) into one single benchmark, which we argue will contribute to faster and more replicable experiments on Twitter. Our second contribution is a suite of experiments that serve two purposes. First, to establish baselines using fine-tuned and ICL approaches, and second, for analysing and producing insights from our experimental results, namely that overall better performance is obtained with fine-tuned masked language models when compared to equally sized text generation architectures, and that zero and few-shot approaches generally struggle as they are not eas-

---

[1]SUPERTWEETEVAL is available at the following link: https://huggingface.co/datasets/cardiffnlp/super_tweeteval

ily to adapt for certain social media tasks. In sum, our results show that SUPERTWEETEVAL is still challenging for general-purpose language models, regardless of their size and domain, and that specialised models are instead more competitive.

## 2 Related Work

The advent of general-purpose NLP systems (namely LMs) has led to a proliferation of unified benchmarks agglutinating diverse NLP tasks. The General Language Understanding Evaluation benchmark (Wang et al., 2018, GLUE) was one of the first efforts to provide a large-scale benchmark composed of diverse tasks such as natural language inference or textual similarity. GLUE was composed of relatively simple and homogeneous tasks, and saw automatic systems quickly reach human performance in some cases (Yang et al., 2019). Because of this, SuperGLUE (Wang et al., 2019) was developed with the same spirit but included a wider range of tasks and settings. Since then, other general-purpose benchmarks for language models, especially those of the new generation, have emerged, such as MMMU (Hendrycks et al., 2021) and BIG-Bench (Srivastava et al., 2022).

In terms of social media research, there are many tasks that require modelling textual content. TweetEval (Barbieri et al., 2020) was the first unified benchmark that agglutinated different tasks into the same benchmark. However, TweetEval is limited to tweet classification, including emotion recognition (Mohammad et al., 2018), emoji prediction (Barbieri et al., 2018a), irony detection (Van Hee et al., 2018), hate speech detection (Basile et al., 2019a), offensive language identification (Zampieri et al., 2019b), sentiment analysis (Rosenthal et al., 2017), and stance detection (Mohammad et al., 2016). Similar to the evolution of GLUE into SuperGLUE, the aim of this paper and SUPERTWEETEVAL is to construct a benchmark that is robust, large, and especially consisting of diverse tasks and settings for social media NLP research, and in particular, Twitter.

## 3 Datasets

SUPERTWEETEVAL includes a variety of Twitter-specific NLP tasks. For each of them, we include a relevant dataset that can be used as a proxy to test the performance of models on that task[2]. In this

---

[2]More details and justifications for the selection of datasets used in this benchmark are provided in Appendix A.1.

section, we describe the datasets used for each task, which we have split into three types: (1) existing datasets that have been included in the benchmark as they are; (2) datasets that have been adapted to suit the needs of the benchmark; and (3) datasets that we have constructed from scratch as part of this paper, which did not exist in previous literature.

### 3.1 Existing Datasets

**Intimacy Analysis (TWEETINTIMACY)** Intimacy is an important social aspect of language communication (Pei and Jurgens, 2020). We use the English subset of the MINT dataset (Pei et al., 2022), which contains 1,983 English tweets annotated with intimacy scores ranging from 1 to 5, with 1 meaning "Not intimate at all" and 5, "Very intimate".

**Meaning-Shift Detection (TEMPOWIC)** This task focuses on the understanding of language evolution through time which, while a popular research topic (Luu et al., 2022; Agarwal and Nenkova, 2022), remains a challenging problem, specifically in Twitter. In SUPERTWEETEVAL, we utilise TEMPOWIC (Loureiro et al., 2022b), a dataset comprised of 3,294 tweets. Here, two tweets from different time periods and a target word are provided, and the goal is to recognise whether the target word's meaning is the same in the two tweets (binary classification).

**Sentiment Classification (TWEETSENTIMENT)** Sentiment analysis has been extensively studied both in general (Medhat et al., 2014; Wankhade et al., 2022) and social media context (Barbieri et al., 2022a; Marcec and Likic, 2022). In SUPERTWEETEVAL, we utilise the data presented in the SemEval 2017 Task 4, subtask C (Rosenthal et al., 2017). The data are formatted as a "Topic Based Sentiment Analysis" task where each tweet is given a sentiment label on a 5-point scale ('strongly negative', 'negative', 'negative or neutral', 'positive', 'strongly positive') regarding a specific target. In total 43,011 tweets and 325 different topics are present.

**Emotion Classification (TWEETEMOTION)** Similar to sentiment analysis, emotion classification has been a popular topic of study (Kušen et al., 2017; He et al., 2016) and has been used to better understand users' behaviours and intentions in social media (Son et al., 2022; Corbett and Savarimuthu, 2022). For our use case, we utilise

the English subset of the 2018 SemEval task 1: *Affect in Tweets*, subtask: *E-c* (Mohammad et al., 2018). A total of 7,168 tweets are present and are labelled with one or more emotions based on their context. The labels are selected from a taxonomy of 11 emotions (plus *neutral* indicating the absence of emotion) such as *anger*, *fear*, and *joy*.[3]

**Topic Classification (TWEETTOPIC)** Topic classification is a method commonly used to perform targeted analysis on social media data. This is a challenging task, due to the ever increasing amount of data produced in social platforms (Weller, 2015; Stieglitz et al., 2018). In SUPER-TWEETEVAL we use the multi-label setting of the TWEETTOPIC dataset (Antypas et al., 2022) for topic classification. The dataset consists of 6,837 tweets that have been assigned one or more topics. The taxonomy of topics used was selected by a team of social media experts from Snap Inc. and consists of 19 broad topics tailored for social media content such as *sports* or *music*.[4]

**Question Answering (TWEETQA)** As a generative task, we consider an abstract question answering (QA) task on Twitter. To this end, we rely on TWEETQA (Xiong et al., 2019), which consists of a tweet and an answer as input, with the answer to the question as the output. Note that the answer may not be explicitly included in the tweet. The dataset contains 9,489/1,086/1,203 tweets for training/validation/test splits, respectively.

**NER (TWEETNER7)** For Name Entity Recognition (NER), we include the TWEETNER7 dataset (Ushio et al., 2022b). This dataset contains 6,837 tweets and seven different labels: *person, location, corporation, creative work, group, product*, while also offering a temporal split (train and test splits stemming from different time periods).

## 3.2 Adapted Datasets

**Named Entity Disambiguation (TWEETNERD)** The original TWEETNERD dataset (Mishra et al., 2022) is a collection of tweets, a target phrase within the tweet and a Wikidata entity ID to which the target phrase refers in the context of tweet. To make the task more accessible for the evaluation of language models, we convert the dataset to a binary classification task: given the tweet, target phrase and a possible definition of the target phrase,

the system's objective is to determine whether the provided definition aligns with the target phrase in the given context (positive) or not (negative).

First, to obtain positive instances with matching definitions, we use the definition provided for the gold Wikidata item ID. Then, we associate a negative instance for each target word's positive instances. For this, a maximum of top 10 candidates were pulled from the Wikidata API by searching for the target phrase of a positive. Then, candidates with low page views were eliminated to remove noise, and negative instances were chosen randomly from the pool of candidate definitions.

**Hate Speech Detection (TWEETHATE)** The presence of hate speech in social media is an ever increasing problem, with hateful content being spread in various online communities (Udanor and Anyanwu, 2019; Walther and McCoy, 2021). We utilise *Measuring Hate Speech* (Sachdeva et al., 2022) which consists of 39,565 social media (YouTube, Reddit, Twitter) manually annotated comments. The coders were asked to annotate each entry on 10 different attributes such as the presence of sentiment, respect, insults, and others; and also indicate the target of the comment (e.g. age, disability). The authors use Rasch measurement theory (Rasch, 1960) to aggregate each annotator's rating for every label in a continuous value which then can be mapped to a binary value.

For our needs, only entries extracted from Twitter were considered. Each tweet was assigned a label if at least two out of five annotators agreed on it. We opted out of a majority rule in order to acquire a dataset that can be used to train models capable of handling real-world, complex data (Mohammad et al., 2018; Antypas et al., 2022). A small amount of tweets with more than one label were discarded. The final dataset contains 7,168 tweets and 8 different labels[5].

**Question Generation (TWEETQG)** By leveraging the TWEETQA dataset, we re-frame it as a question generation (QG) task, where we use the tweet and the answer as the model input, while the question is the output.

## 3.3 New Datasets

In addition to the previous datasets that were directly integrated into the benchmark with minimal preprocessing or adapted from existing ones,

---

[3]Full list of emotions can be found in Appendix: Table 6.
[4]Full list of topics can be found in Appendix: Table 7.

[5]Full list of labels can be found in Appendix: Table 8.

we also constructed two additional datasets from scratch that complement the initial list of tasks and datasets. These are emoji prediction over 100 labels (Section 3.3.1) and tweet similarity (Section 3.3.2).

### 3.3.1 Emoji Prediction (TWEETEMOJI100)

This task aims to expand on previous emoji classification problems with 20 classes (Barbieri et al., 2017b, 2018b) by introducing a more challenging dataset with 100 different emojis (TWEETEMOJI100). TWEETEMOJI100 consists of a more recent corpus, an important feature to consider in the ever evolving setting of social media, and takes into account a wider variety of emojis. TWEETEMOJI100 considers tweets with only one emoji present at the end of the text which is removed and used as our label to create a multi-class classification setting.

For the creation if the dataset, an existing large collection of tweets (37,083,959) (Loureiro et al., 2022a) is taken into account to extract the hundred most frequent emoji. For each emoji selected, we collected 500 tweets every day for the time period between 01-01-2020 to 01-01-2023. In total 7,379,453 new tweets were gathered through the Twitter API utilising the *Twarc* library (Summers, 2013).

Following tweet collection, we filtered all entries that contained more than one emoji and entries where the emoji was not present at the end of the tweet. To avoid highly similar tweets that may have different emojis, we also removed near duplicated entries. This is accomplished by applying a normalisation step where (1) URLs and mentions are removed, and (2) entries that are considered duplicated based on their lemmatised form are ignored. Finally, colour variations of the heart, circle, and square emoji were ignored. All emojis present in TWEETEMOJI100 and their distribution can be found in Figure 1 of the Appendix.

### 3.3.2 Tweet Similarity (TWEETSIM)

Given the importance of textual similarity dataset in NLP (Cer et al., 2017) and the lack of such datasets in social media, we decided to construct a new dataset focused on tweet similarity. Given two tweets as input, the tweet similarity task consists of assigning them a 0 to 5 score according to their similarity.

**Sampling** Similarly to the TEMPOWIC tweet sampling procedure, we followed the approach of Chen et al. (2021) to detect trending hashtags for the period between 2020 and 2021, based on the corpora collected for TimeLM (Loureiro et al., 2022a). Afterwards, we randomly selected a diverse set of hashtags and collected an additional sample of tweets featuring those hashtags (i.e., most common hashtag only appears on 25 pairs). The resulting dataset features 1,000 tweet pairs, with the inclusion of 20% randomly paired tweets for completeness.

**Annotation** All the tweet pairs were then rated with by Likert-like scale by three independent annotators[6]. All the annotators were native English speakers and were paid fairly through our institutional student job provider[7]. The final inter-annotator, as measured by annotator pairwise Spearman correlation, was 0.70.

## 4 SUPERTWEETEVAL: The Unified Benchmark

We convert all datasets presented in the previous section to the same JSON format, unifying them with the same notation, preprocessing and style. Table 1 provides an overview of each task and dataset while also providing example entries.

### 4.1 Preprocessing

A common preprocessing pipeline is applied to all the collected datasets aiming to standardise them and provide a uniform and easy-to-use format. Firstly, all URL links are masked as *{URL}*, both for privacy reasons, and for concentrating the focus of our tasks to the main text context of each tweet. Furthermore, all mentions of non-verified users are masked with *@user* to maintain their anonymity. Finally, an attempt is made to unify features and label/score naming to increase the datasets' ease-of-use.

### 4.2 Evaluation Metrics

To unify evaluation metrics for a better understandability, we selected and converted all metrics in a percentage-based 0-100 scale, in which higher scores represent better model predictions.

**TWEETSENTIMENT** *Macro Averaged Mean Absolute Error* ($MAE^M$) (Baccianella et al., 2009)

---

[6]Guidelines are available in Appendix A.3

[7]To avoid breaking anonymity rules, more details about the annotators and compensation will be provided upon acceptance.

| Task (Dataset) | Example Input | Example Output |
|---|---|---|
| NER (TWEETNER7) | **Tweet**: Winter solstice 2019 : A short day that 's long on ancient traditions url via @CNN_Travel | Winter solstice 2019: event @CNN_Travel: product |
| Emotion Classification (TWEETEMOTION) | **Tweet**: Whatever you decide to do make sure it makes you #happy. | joy, love, optimism |
| Question Generation (TWEETQG) | **Tweet**: 5 years in 5 seconds. Darren Booth (@darbooth) January 25, 2013 **Context**: vine | what site does the link take you to? |
| Name Entity Disambiguation (TWEETNERD) | **Tweet**: hella excited for ios 15 because siri reads notifications out loud to you [...] **Target**: siri **Definition**: intelligent personal assistant on various Apple devices | True |
| Sentiment Classification (TWEETSENTIMENT) | **Tweet**: #ArianaGrande Ari By Ariana Grande 80% Full url #Singer #Actress url **Target**: #ArianaGrande | negative or neutral |
| Meaning Shift Detection (TEMPOWIC) | **Tweet 1**: The minute I can walk well I'm going to delta pot **Tweet 2**: Then this new delta variant out im vaccinated but stilllll likeee' **Target**: delta | False |
| Emoji Classification (TWEETEMOJI100) | **Tweet**: SpiderMAtS back at it | 🔥 |
| Intimacy Analysis (TWEETINTIMACY) | **Tweet**: @user SKY scored 4 less runs just lol | 1.20 |
| Question Answering (TWEETQA) | **Tweet**: 5 years in 5 seconds. Darren Booth (@user) January 25, 2013 **Question**: which measurements of time are mentioned? | years and seconds |
| Topic Classification (TWEETTOPIC) | **Tweet**: Sweet, #IOWAvsISU is a nationally televised night game! Nebraska getting bumped to @FOX_Business is just a bonus. | film_tv_&_video, sports |
| Hate Speech Detection (TWEETHATE) | **Tweet**: Support Black Trans youth url | not_hate |
| Tweet Similarity (TWEETSIM) | **Tweet 1**: I wish kayvee all the best #bbnaija **Tweet 2**: Sammie about to cry to the housemates all night #bbnaija | 2.33 |

Table 1: Example input and output for each and dataset included in SUPERTWEETEVAL.

is selected as the evaluation metric for the Sentiment Classification task. To better integrate it in our benchmark, we use $1 - MAE^M$ as our score and cap the negative values to 0. In contrast to F1-scores, $MAE^M$ (also used in the original SemEval competition) takes into account the order of the labels and provides a better understanding of the performance of the models.

**TWEETEMOTION, TWEETTOPIC and TWEETNER7** For the multi-label classification tasks of TWEETEMOTION and *Topic Classification* the standard *average macro-F1* score is used. Metrics like *Accuracy* score, and *Jaccard Index* were initially considered, however, macro-F1 will encourage the development of more precise and accurate models across classes. *Average macro-F1* is also used for the NER task (similar to the TWEETNER7 original paper).

**TWEETEMOJI100** Considering the large number of labels present in the Emoji Classification task and the fact that some of the emojis can be a close match for more than one entry, *Accuracy at top 5* is selected as the official evaluation metric.

**TWEETSIM & TWEETINTIMACY** For both regression tasks, Spearman's correlation *r* is used as the main evaluation metric. This metric focuses on the relationship between predicted and actual ranks instead of the absolute errors between them (i.e. Mean Absolute Error). Negative values are also capped to 0.

**TWEETNERD & TEMPOWIC** For both binary classification tasks we use *Accuracy* score. The classes in both datasets are relatively balanced (fully balanced in the case of TWEETNERD) and thus Accuracy provides a reliable metric.

**TWEETHATE** In this task we utilise a combination of micro and macro *F1* scores as an evaluation metric. Specifically, we first group all entries classified as hate speech as one class, and together with the not-hate class the micro-F1 score is calculated. Then, the macro-F1 for only the hate speech sub-classes is calculated. Finally, we report the average of these two scores. This "combined F1" score is selected because: (1) it weights heavily the most important decision (a tweet being hateful or not) and (2) does not penalise unfairly poor performance in low frequency hate speech sub-classes.

| Task (Dataset) | Train | Valid. | Test |
|---|---|---|---|
| TWEETNER7 | 4,616 | 576 | 2,807 |
| TWEETEMOTION | 6,838 | 886 | 3,259 |
| TWEETQG | 9,489 | 1,086 | 1,203 |
| TWEETNERD | 20,164 | 4,100 | 20,075 |
| TWEETSENTIMENT | 26,632 | 4,000 | 12,379 |
| TEMPOWIC | 1,427 | 395 | 1,472 |
| TWEETEMOJI100 | 50,000 | 5,000 | 50,000 |
| TWEETINTIMACY | 1,191 | 396 | 396 |
| TWEETQA | 9,489 | 1,086 | 1,203 |
| TWEETTOPIC | 4,585 | 573 | 1,679 |
| TWEETHATE | 5,019 | 716 | 1,433 |
| TWEETSIM | 450 | 100 | 450 |

Table 2: Number of tweets in the train, validation (Valid.) and test splits for each of the tasks in SUPER-TWEETEVAL.

**TWEETQA & TWEETQG**   For the evaluation metrics of generative tasks, we employ the answer-span F1 score for TWEETQA following Rajpurkar et al. (2016), and METEOR (Denkowski and Lavie, 2014) for TWEETQG, which has been shown to be a well-correlated metric for QG (Ushio et al., 2022a).

### 4.3   Statistics

Overall, SUPERTWEETEVAL consists of 255,170 tweets across twelve different datasets. For each task, we consider the training/validation/test splits as presented in their resource paper (or in the model released by the authors of the papers). Exceptions are the TWEETHATE, TWEETSIM and TWEETE-MOJI100 tasks where new data splits were created. Table 2 displays the final distribution of tweets in each split for each task.

## 5   Experimental Setting

For the evaluation, we rely on the datasets and splits presented in Section 4.3. In particular, we evaluate all models on the test splits. Each dataset uses a different evaluation metric, as introduced in Section 4.2.

### 5.1   Naive Baselines

To establish a lower performance threshold for each task, naive baselines are also included. For the classification tasks (TWEETEMOTION, TWEET-NERD, TEMPOWIC, TWEETTOPIC, TWEET-HATE, TWEETEMOJI100) a *Majority* classifier (most frequent class in training) is employed.

For the regression tasks, the naive model always outputs the average value of the training set (TWEETINTIMACY, TWEETSIM), and for Sentiment Classification (ordinal classification) the output is always 'negative or neutral'. Finally, for the text generation tasks of QA & QG our naive model always returns the input text, and for the NER task it outputs random tokens assigned with random entities.

### 5.2   Fine-tuning

**Model Selection**   For the fine-tuning setting we consider eight different models[8]: OPT (Zhang et al., 2022a), FlanT5 (Chung et al., 2022), RoBERTa (Liu et al., 2019), and TimeLM (Loureiro et al., 2022a). The selection of the models was done based on: (1) their relatively small size, with the smallest model having 85 million parameters (FlanT5$_{SMALL}$) and the largest 354 million (RoBERTa$_{LARGE}$). (2) The architectures of the models and training process. FlanT5 is an encoder-decoder model trained with a text-to-text format; OPT is a decoder only model and its training corpus includes a large portion of Reddit data; and finally, RoBERTa, a traditional masked language model, and TimeLM which are based on the same architecture, but their training corpus (specialised on social media, and particularly Twitter) makes them an interesting candidate. For the NER task, only the results from the RoBERTa based models are reported since adapting OPT and FlanT5 for this task is not trivial, and hence this endeavour is left for future work.

**Training**   The implementations provided by HuggingFace (Wolf et al., 2020) are used to train and evaluate all language models, while we utilise Ray Tune (Liaw et al., 2018) for optimising the number of epochs, learning rate, warmup steps, and batch size hyper-parameters. The hyper-parameter optimisation is done by performing a random search over 10 different runs for each model.

### 5.3   Zero & Few Shot

Further to the *fine-tune* experimental setting, two in-context learning settings are established: zero and few shot. Aiming to explore the challenges that arise when testing SUPERTWEETEVAL in such settings, we select the larger versions of the

---

[8]Details of the model can be found in the Appendix: Table 9.

| | FlanT5$_{\text{SMALL}}$ | FlanT5$_{\text{BASE}}$ | OPT$_{125M}$ | OPT$_{350M}$ | RoBERTa$_{\text{BASE}}$ | RoBERTa$_{\text{LARGE}}$ | TimeLM$_{\text{BASE}}$ | TimeLM$_{\text{Large}}$ | Naive |
|---|---|---|---|---|---|---|---|---|---|
| TEMPOWIC | 63.59 | 62.84 | 58.02 | 67.66 | 65.08 | 63.86 | 68.14 | **68.41** | 63.45 |
| TWEETEMOJI100 | 1.77 | 0.75 | 32.18 | 31.51 | 31.56 | 34.09 | 33.45 | **35.64** | 8.55 |
| TWEETEMOTION | 37.49 | 47.73 | 54.62 | 55.36 | 55.27 | 57.95 | 55.61 | **58.53** | 4.58 |
| TWEETHATE | 55.01 | 69.48 | 76.31 | 76.67 | 71.09 | 82.32 | 79.38 | **82.54** | 35.45 |
| TWEETINTIMACY | 45.92 | 56.90 | 45.83 | 41.06 | 52.44 | 22.28 | **68.95** | 58.53 | 3.73 |
| TWEETNER7 | - | - | - | - | 59.10 | 60.00 | 58.20 | **60.40** | 2.12 |
| TWEETNERD | 76.70 | 54.79 | 83.04 | 84.11 | 83.19 | 84.92 | 83.95 | **85.30** | 50.00 |
| TWEETQA | 53.64 | **66.09** | 19.60 | 17.43 | - | - | - | - | 12.70 |
| TWEETQG | 10.73 | **44.04** | 3.97 | 3.79 | - | - | - | - | 25.36 |
| TWEETSENTIMENT | 6.98 | 2.38 | 48.53 | 49.29 | 50.75 | 54.50 | 51.89 | **54.65** | 0.00 |
| TWEETSIM | 9.70 | 5.45 | 65.99 | 66.41 | 74.42 | 68.15 | 72.24 | **74.64** | 0.00 |
| TWEETTOPIC | 23.16 | 40.53 | 55.65 | 58.78 | 45.34 | 58.71 | 36.65 | **58.84** | 2.30 |

Table 3: SUPERTWEETEVAL individual task results of selected models in the fine-tuning setting.

FlanT5 models (FlanT5$_{\text{XL}}$ & FlanT5$_{\text{XXL}}$), OPT-IML$_{1.3B}$[9] and also `text-ada-001` from OpenAI, a small version of GPT-3 (Brown et al., 2020), and `chat-gpt-3.5-turbo`[10], and are evaluated in each task.

In both settings, we prompt the models three times and report the average result of the runs. Specifically, in the few-shot setting we sample different examples extracted from the validation set of each task for each run. The number of examples sampled for few-shot was based on the given task. For regression and text generation tasks, we provide five random examples in each prompt, while for classification tasks we include one example per class with a maximum of five examples. The prompts used in our experiments are, in their majority, based on the instructions used in the FlanT5 (Chung et al., 2022), and OPT-IML (Iyer et al., 2023) papers[11].

As a final note, we forfeit reporting the results of zero/few-shot settings on TWEETNER7 and TWEETEMOJI100 as our initial experiments were unsuccessful. This is mainly due to: (1) limitations of the models themselves (e.g. FlanT5 models are not trained with emojis); (2) evaluation difficulties (TWEETEMOJI100 is evaluated using Accuracy at top 5 which leads to complications on the few-shot setting as only one emoji is included in the gold standard); and (3) issues that arose with the prompts tested (see Section C in the Appendix).

## 6 Results

**Fine-tuning** The results from the fine-tuning setting (Table 3) provide an indication of the level

of difficulty of each task. Not surprisingly, most models seem to perform relatively well on simpler datasets such as TWEETHATE (best: 0.8254) and TWEETNERD (best: 0.8530). However, the majority of the tasks appear to still offer an adequate challenge, with the best performing overall model (TimeLM$_{\text{LARGE}}$) achieving an average score of 0.63 across all tasks tested. Specifically, the most difficult tasks appear to be TWEETEMOJI100 and TWEETQG, where all models perform below 0.5.

Finally, regarding the models' architectures tested, our results indicate that the RoBERTa models (and specifically TimeLM$_{\text{LARGE}}$) display a better performance across most tasks when compared to FlanT5 and OPT counterparts.

**Zero & few shot** When considering the results of our zero/few shot experiments (Table 4), a clear trend is revealed where most models tested fail to achieve better, or even similar, results to those that were fine-tuned. Exception to this is `chat-gpt-3.5-turbo` which in some tasks such as TEMPOWIC and TWEETNERD achieves similar, but still lower, performance to the fine-tuned models, while it also achieves the best score in the TWEETQA. However, its performance must be viewed with caution as due to its closed nature as there is a possibility that the GPT models may have already been trained on the datasets collected (including test splits) providing them an unfair advantage.

The difference in performance, particularly in the regression and ordinal classification tasks of TWEETINTIMACY and TWEETSENTIMENT, is significant. The best performing model, FlanT5$_{\text{XXL}}$, achieves, in a few-shot setting, scores of 29.96 and 25.72 in TWEETINTIMACY and TWEETSENTIMENT respectively, which is more than a 50% drop

---

[9]The IML version (Iyer et al., 2023) is selected as it is trained in a similar way to FlanT5.

[10]https://openai.com/chatgpt

[11]Detailed prompts for each task can be found in Appendix C.

| | Model | Zero-shot | Few-shot |
|---|---|---|---|
| **TempoWic** | FlanT5$_{XL}$ | 58.22 | **66.33** |
| | FlanT5$_{XXL}$ | 38.09 | 65.71 |
| | OPT-IML$_{1.3B}$ | **62.14** | 60.33 |
| | text-ada-001 | 33.26 | 39.29 |
| | chat-gpt-3.5-turbo* | 64.95 | 68.34 |
| **TweetEmo.** | FlanT5$_{XL}$ | 30.08 | 30.19 |
| | FlanT5$_{XXL}$ | **32.77** | **36.63** |
| | OPT-IML$_{1.3B}$ | 19.96 | 24.66 |
| | text-ada-001 | 0.42 | 20.06 |
| | chat-gpt-3.5-turbo* | 45.17 | 51.56 |
| **TweetHate** | FlanT5$_{XL}$ | **51.05** | **54.87** |
| | FlanT5$_{XXL}$ | 41.37 | 43.74 |
| | OPT-IML$_{1.3B}$ | 28.46 | 23.10 |
| | text-ada-001 | 35.45 | 29.94 |
| | chat-gpt-3.5-turbo* | 63.42 | 57.9 |
| **TweetInt.** | FlanT5$_{XL}$ | 28.22 | 28.98 |
| | FlanT5$_{XXL}$ | **29.96** | **29.68** |
| | OPT-IML$_{1.3B}$ | 0.00 | 1.94 |
| | text-ada-001 | 3.17 | 0.92 |
| | chat-gpt-3.5-turbo* | 41.82 | 53.17 |
| **TweetQA** | FlanT5$_{XL}$ | 53.85 | 54.33 |
| | FlanT5$_{XXL}$ | **53.88** | **64.44** |
| | OPT-IML$_{1.3B}$ | 50.85 | 53.50 |
| | text-ada-001 | 17.99 | 17.99 |
| | chat-gpt-3.5-turbo* | 32.90 | 70.51 |

| | Model | Zero-shot | Few-shot |
|---|---|---|---|
| **TweetNERD** | FlanT5$_{XL}$ | 67.25 | 67.54 |
| | FlanT5$_{XXL}$ | **75.56** | **73.94** |
| | OPT-IML$_{1.3B}$ | 54.71 | 52.80 |
| | text-ada-001 | 35.09 | 50.02 |
| | chat-gpt-3.5-turbo* | 69.97 | 80.28 |
| **TweetQG** | FlanT5$_{XL}$ | **21.76** | **22.15** |
| | FlanT5$_{XXL}$ | 21.05 | 21.87 |
| | OPT-IML$_{1.3B}$ | 20.34 | 14.51 |
| | text-ada-001 | 19.09 | 14.49 |
| | chat-gpt-3.5-turbo* | 23.25 | 33.3 |
| **TweetSent.** | FlanT5$_{XL}$ | 0.50 | 6.44 |
| | FlanT5$_{XXL}$ | **28.30** | **25.72** |
| | OPT-IML$_{1.3B}$ | 18.84 | 11.09 |
| | text-ada-001 | 0.00 | 0.00 |
| | chat-gpt-3.5-turbo* | 42.99 | 41.22 |
| **TweetSim** | FlanT5$_{XL}$ | **59.46** | 54.29 |
| | FlanT5$_{XXL}$ | 56.76 | **61.69** |
| | OPT-IML$_{1.3B}$ | 43.91 | 9.92 |
| | text-ada-001 | 0.00 | 8.71 |
| | chat-gpt-3.5-turbo* | 68.94 | 57.74 |
| **TweetTopic** | FlanT5$_{XL}$ | 36.16 | 34.73 |
| | FlanT5$_{XXL}$ | **36.25** | **37.35** |
| | OPT-IML$_{1.3B}$ | 13.43 | 8.59 |
| | text-ada-001 | 0.00 | 4.16 |
| | chat-gpt-3.5-turbo* | 54.77 | 48.31 |

Table 4: SUPERTWEETEVAL zero & few shot results. Best results for each task and setting are bolded. Chat-GPT results (marked with *) are included for completeness. We refrained from highlighting ChatGPT results due to its closed and irreproducible nature, as well as the possibility to have been directly trained on some of the test sets.

in performance compared to the scores achieved by the best performing fine-tuned model (68.95 and 54.65).[12]

## 7 Analysis

Aiming to acquire a better understanding of capabilities for the model, and also the challenges that the tasks present, we organise the tasks in smaller *sub-benchmarks* or *clusters* that are aimed at testing a particular feature, and investigate their performance. The clusters defined are created based on the nature of each task as well as the features that are present in them.

**Temporal**[13]. For this cluster, all the datasets that feature a temporal aspect are grouped together. In particular, we include those datasets

that contain data splits from different time periods (TWEETNER7, TEMPOWIC, TWEETTOPIC, and TWEETNERD).

**Multi-label.** In this cluster we include the TWEETTOPIC and TWEETEMOTION datasets, analysing the models' performance in multi-label classification.

**Multi-class.** Similar to the previous cluster, we consider the TWEETSENTIMENT and TWEET-HATE datasets to evaluate the models in single-label multi-class tweet classification.

**Regression.** For this cluster, we include the two regression tasks of TWEETSIM and TWEETINTI-MACY, and also consider TWEETSENTIMENT (ordinal classification).

**Target-based.** We group all datasets that provide information regarding a target word or entity that is used in models' predictions (TWEETSENTIMENT, TEMPOWIC and TWEETNERD).

---

[12]The goal of this paper is not to have the strongest models, but rather to evaluate models out-of-the-box. As such, there are tasks such as TWEETEMOJI100 that are not easily solved by the selected zero/few shot models.

[13]Due to the lack of zero/few-shot results for TWEETNER7, we added the subset *temporal** that does not include NER.

| Mode | Model | Big-label | Disamb. | Generation | Multi-class | Multi-label | Regression | Target | Temporal | Temporal* |
|------|-------|-----------|---------|------------|-------------|-------------|------------|--------|----------|-----------|
| Zero-shot | FlanT5$_{XXL}$ | - | 56.82 | 37.46 | 34.83 | 34.51 | 38.34 | 47.31 | - | 49.97 |
| | OPT-IML$_{1.3B}$ | - | 58.42 | 35.59 | 23.65 | 16.70 | 21.38 | 45.23 | - | 43.42 |
| | chat-gpt-3.5-turbo | - | 67.46 | 32.47 | 53.20 | 49.97 | 51.25 | 59.30 | - | 63.23 |
| Few-shot | FlanT5$_{XXL}$ | - | 69.83 | 43.16 | 34.73 | 36.99 | 39.03 | 55.12 | - | 59.00 |
| | OPT-IML$_{1.3B}$ | - | 56.56 | 34.01 | 17.09 | 16.62 | 8.09 | 41.41 | - | 40.57 |
| | chat-gpt-3.5-turbo | - | 74.31 | 51.90 | 49.56 | 49.93 | 50.71 | 63.28 | - | 65.64 |
| F. tuning | FlanT5$_{BASE}$ | 20.64 | 58.82 | **55.06** | 35.93 | 44.13 | 21.58 | 40.00 | 47.59 | 52.72 |
| | OPT$_{350M}$ | 45.15 | 75.89 | 10.61 | 62.98 | 57.07 | 52.26 | 67.02 | 55.62 | 70.18 |
| | TimeLM$_{LARGE}$ | **47.24** | **76.86** | - | **68.59** | **58.68** | **62.61** | **69.45** | **67.78** | **70.85** |

Table 5: Aggregated results over each test cluster of the best zero-shot, few-shot and fine-tuning methods.

**Big-label.** In this setting we include classification tasks (both multi and single label) that contain a high number of labels (TWEETEMOJI100 with 100 labels and TWEETTOPIC with 19 labels).

**Generation.** TWEETQA and TWEETQG were grouped together to create a setting for evaluating the generation capabilities of the models.

**Disambiguation.** As a final cluster we consider the tasks TEMPOWIC and TWEETNERD which share the goal of understanding and differentiating the meaning of a term between two contexts.

For this analysis, we selected the two best performing models in zero and few shot settings (FlanT5$_{XXL}$, OPT-IML$_{1.3B}$) along with chat-gpt-3.5-turbo, and the best model of each architecture from the fine-tuning experiment (TimeLM$_{LARGE}$, FlanT5$_{BASE}$, and OPT$_{350M}$). Table 5 displays the results of each cluster. Although the comparison across clusters is not straightforward given the different evaluation metrics, the most challenging settings for all model tested appear to be the *Big-label* and *Multi-label* clusters where no score greater than 60 is achieved. Finally, the results again highlight that in-context learning models (both zero- and few-shot) generally underperform compared to smaller models fine-tuned on the full training set, with even ChatGPT failing to attain the best score in any of the test clusters. In general, the fine-tuned TimeLM$_{LARGE}$ model achieve the best results across all non-generation clusters, with the fine-tuned FLAN-T5 model achieving the best results on generation tasks.

## 8 Conclusion

In this paper, we have presented a new social media NLP benchmark, SUPERTWEETEVAL. It goes beyond simple tweet classification, including generative, sequence prediction, and regression tasks, in addition to challenging classification tasks. This benchmark is aimed at unifying Twitter-based evaluation protocols and provides a realistic assessment of models in this difficult and important domain. Our evaluation highlighted the challenging nature of the benchmark and the social media domain. In particular, the results show how recent LLMs struggle with the specialised nature of this domain, with smaller but fine-tuned and more specialised models being more competitive overall. In addition to its evaluation purposes, it can also be used as the basis for multi-task learning, as well as for making use of already-trained models.

## Limitations

The main issue of SUPERTWEETEVAL is its lack of language variety, as it focuses on English only. By making this first English benchmark publicly available, we hope this can pave the way for future research to extend the benchmark for other languages.

In terms of data, the benchmark only includes one dataset per task. We took this choice to both (1) make the benchmark simpler to understand, and (2) because for most tasks only a single dataset was available. As such, conclusions for individual tasks can be limiting. Also, for some tasks, it is hard to know the performance ceiling for models, often reported as human performance. While we could have attempted to provide an upper bound, we believe this is a challenging problem in itself, as also human performance estimates are often unreliable as a performance indicator (Tedeschi et al., 2023).

Finally, our evaluation is focused on a simple setting comparing language models in supervised and zero/few shot settings and only focused on a limited set of LLMs. We did not intend to provide a new model performing well on all tasks, but rather an assessment of current models in similar

conditions. Because of this, we may have not provided the models with their optimal settings. We will release all the evaluation scripts so that other researchers can easily evaluate their model performance on SUPERTWEETEVAL.

## Ethics Statement

Our work aims to contribute and extend research in social media and particularly on Twitter. We propose a unified benchmark that can be utilised to train and evaluate new social media models. The datasets collected in SUPERTWEETEVAL are under public licences and follow the rules of Twitter API. Moreover, given that the data presented includes user generated content we are committed to respecting the privacy of the users. This is achieved by applying the necessary preprossessing to remove user mentions (from non-verified users) and URLs that can be used to identify individuals. We also ensured that none of the dataset splits contain more than 50,000 tweets. Finally, regarding the annotation of TWEETSIM, a fair payment was ensured to be paid for the annotators that took part in its collection.

## Acknowledgements

Jose Camacho-Collados is supported by a UKRI Future Leaders Fellowship. We would like to thank Fangyu Liu and Daniel Loureiro for their involvement in specific tasks in the early stages of this project.

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

## A Datasets

### A.1 Dataset Selection

All the datasets used in this benchmark were carefully selected based on their difficulty level, coverage they offer, and availability (licence). Below we discuss the selection process for tasks that a large selection of pre-existing datasets exists.

**TWEETHATE** For this task we considered a list of 13 different hate speech detection datasets on Twitter (Sachdeva et al., 2022; Samory et al., 2021; Grimminger and Klinger, 2021; Mathew et al., 2021; Zampieri et al., 2019a; Davidson et al., 2017; Waseem and Hovy, 2016; Waseem, 2016; Ousidhoum et al., 2019; Basile et al., 2019b; Mandl et al., 2019; Vidgen et al., 2020; Founta et al., 2018). The final dataset contains five distinct hate-speech classes, in contrast to other available options, e.g. two hate classes present in SemEval's 2019 task 5 (Basile et al., 2019b), and also utilises a more clear taxonomy than other available datasets (e.g. HateXplain, Mathew et al. AAAI, 2021 (Mathew et al., 2021)).

**TWEETSENTIMENT** We opted out of using datasets that consider sentiment analysis as a classification (Rane and Kumar, 2018) or regression (Saad and Yang, 2019) task and instead use TWEETSENTIMENT(aspect based on a five point scale) as

a more challenging setting for more recent models and architectures. Finally, as it was part of a SemEval competition TWEETSENTIMENTis already well-known and tested dataset.

**TWEETEMOTION** Similar for the Sentiment Analysis task, the dataset pool for the TWEETEMOTION task was rather limited when considering our needs (multi-label classification & Twitter based) (Balabantaray et al., 2012; Strapparava and Mihalcea, 2007). Again the data selected is well tested SemEval dataset that fits the needs of our task.

**TWEETNER7** Finally, the TWEETNER7 dataset was selected instead of similar ones (Rijhwani and Preotiuc-Pietro, 2020; Jiang et al., 2022; Derczynski et al., 2017; **?**) as it provides a larger taxonomy of entities, a larger dataset with a uniformed distribution, and temporal characteristics (a more recent corpus) which are appreciated for building the sub-clusters used.

### A.2 Detailed Classes

| | |
|---|---|
| 1. anger | 8. pessimism |
| 2. anticipation | 9. sadness |
| 3. disgust | 10. surprise |
| 4. fear | 11. trust |
| 5. joy | 12. neutral (no emotion) |
| 6. love | |
| 7. optimism | |

Table 6: Emotions present in TWEETEMOTION.

| | |
|---|---|
| 1. arts & culture | 12. music |
| 2. business & entrepreneurs | 13. news & social concern |
| 3. celebrity & pop culture | 14. other hobbies |
| 4. diaries & daily life | 15. relationships |
| 5. famliy | 16. science & technology |
| 6. fashion & style | 17. sports |
| 7. film tv & video | 18. travel & adventure |
| 8. fitness & dining | 19. youth & student life |
| 9. food & dining | |
| 10. gaming | |

Table 7: Topics present in TWEETTOPIC.

| | |
|---|---|
| 1. hate_gender | 5. hate_origin |
| 2. hate_race | 6. hate_disability |
| 3. hate_sexuality | 7. hate_age |
| 4. hate_religion | 8. not_hate. |

Table 8: Classes present in TWEETHATE.

|  | gold_label | train | validation | test | gold_label | train | validation | test |
|---|---|---|---|---|---|---|---|---|
| 0 | (emoji) | 913 | 91 | 912 | (emoji) | 521 | 52 | 521 |
| 1 | (emoji) | 900 | 90 | 900 | (emoji) | 518 | 52 | 519 |
| 2 | (emoji) | 827 | 83 | 827 | (emoji) | 517 | 52 | 518 |
| 3 | (emoji) | 827 | 83 | 827 | (emoji) | 514 | 51 | 514 |
| 4 | (emoji) | 813 | 81 | 812 | (emoji) | 504 | 50 | 503 |
| 5 | (emoji) | 812 | 81 | 812 | (emoji) | 497 | 50 | 497 |
| 6 | (emoji) | 796 | 80 | 796 | (emoji) | 493 | 49 | 492 |
| 7 | (emoji) | 791 | 79 | 791 | (emoji) | 483 | 48 | 483 |
| 8 | (emoji) | 790 | 79 | 790 | (emoji) | 461 | 46 | 460 |
| 9 | (emoji) | 787 | 79 | 787 | (emoji) | 460 | 46 | 460 |
| 10 | (emoji) | 772 | 77 | 772 | (emoji) | 456 | 46 | 457 |
| 11 | (emoji) | 757 | 76 | 758 | (emoji) | 448 | 45 | 448 |
| 12 | (emoji) | 751 | 75 | 750 | (emoji) | 444 | 44 | 443 |
| 13 | (emoji) | 746 | 75 | 747 | (emoji) | 436 | 44 | 437 |
| 14 | (emoji) | 729 | 73 | 729 | (emoji) | 424 | 42 | 424 |
| 15 | (emoji) | 727 | 73 | 728 | (emoji) | 418 | 42 | 418 |
| 16 | (emoji) | 726 | 73 | 727 | (emoji) | 414 | 41 | 414 |
| 17 | (emoji) | 725 | 72 | 725 | (emoji) | 387 | 39 | 387 |
| 18 | (emoji) | 718 | 72 | 719 | (emoji) | 382 | 38 | 382 |
| 19 | (emoji) | 715 | 71 | 715 | (emoji) | 370 | 37 | 370 |
| 20 | (emoji) | 705 | 70 | 704 | (emoji) | 355 | 36 | 355 |
| 21 | (emoji) | 697 | 70 | 697 | (emoji) | 355 | 36 | 355 |
| 22 | (emoji) | 692 | 69 | 692 | (emoji) | 347 | 35 | 348 |
| 23 | (emoji) | 692 | 69 | 691 | (emoji) | 340 | 34 | 340 |
| 24 | (emoji) | 687 | 69 | 688 | (emoji) | 337 | 34 | 337 |
| 25 | (emoji) | 680 | 68 | 679 | (emoji) | 331 | 33 | 330 |
| 26 | (emoji) | 679 | 68 | 679 | (emoji) | 328 | 33 | 329 |
| 27 | (emoji) | 668 | 67 | 669 | (emoji) | 324 | 32 | 323 |
| 28 | (emoji) | 667 | 67 | 667 | (emoji) | 308 | 31 | 309 |
| 29 | (emoji) | 645 | 65 | 645 | (emoji) | 287 | 29 | 287 |
| 30 | (emoji) | 636 | 63 | 636 | (emoji) | 285 | 29 | 286 |
| 31 | (emoji) | 615 | 62 | 616 | (emoji) | 278 | 28 | 279 |
| 32 | (emoji) | 610 | 61 | 609 | (emoji) | 271 | 27 | 270 |
| 33 | (emoji) | 609 | 61 | 609 | (emoji) | 257 | 26 | 258 |
| 34 | (emoji) | 604 | 60 | 603 | (emoji) | 250 | 25 | 250 |
| 35 | (emoji) | 599 | 60 | 598 | (emoji) | 245 | 24 | 245 |
| 36 | (emoji) | 597 | 60 | 598 | (emoji) | 225 | 22 | 225 |
| 37 | (emoji) | 595 | 59 | 595 | (emoji) | 225 | 23 | 226 |
| 38 | (emoji) | 593 | 59 | 593 | (emoji) | 220 | 22 | 219 |
| 39 | (emoji) | 592 | 59 | 591 | (emoji) | 213 | 21 | 213 |
| 40 | (emoji) | 589 | 59 | 588 | (emoji) | 195 | 20 | 196 |
| 41 | (emoji) | 584 | 58 | 584 | (emoji) | 181 | 18 | 181 |
| 42 | (emoji) | 562 | 56 | 562 | (emoji) | 165 | 16 | 164 |
| 43 | (emoji) | 555 | 55 | 554 | (emoji) | 152 | 15 | 151 |
| 44 | (emoji) | 545 | 55 | 545 | (emoji) | 150 | 15 | 150 |
| 45 | (emoji) | 545 | 54 | 545 | (emoji) | 61 | 6 | 61 |
| 46 | (emoji) | 534 | 53 | 534 | (emoji) | 50 | 5 | 50 |
| 47 | (emoji) | 525 | 53 | 526 | (emoji) | 50 | 5 | 50 |
| 48 | (emoji) | 524 | 52 | 524 | (emoji) | 50 | 5 | 50 |
| 49 | (emoji) | 521 | 52 | 521 | (emoji) | 50 | 5 | 50 |

Figure 1: Emojis distribution for each split.

## A.3 Annotator Guidelines of TWEETSIM

The dataset will be composed of pairs of tweets and a relatedness score. The annotation task will, therefore, consists of scoring how related or similar two tweets are according to the following scale:

(5) Tweets are equivalent, even if some minor de-

tails may differ (e.g., commenting about the same situation in different ways, one being more complete than the other, etc.)

**(4)** Almost equivalent, refers to the same situation/event/person but with possibly relevant differences, such as missing significant details.

**(3)** Not equivalent, but shares details about a similar situation/event/person. Could be tweets around a similar event but with a different emotion or sentiment towards it.

**(2)** Categorically related, tweets are on the same topic or category (e.g. sports, politics).

**(1)** Loosely related, there is something minor in common (e.g. same energy/sentiment, same type, etc.).

**(0)** No relation, the tweets do not have anything in common.

Please note that only exact numbers should be used (e.g. 2 or 3) without any decimal.

## B  Models Details

## C  Zero-shot Prompts

We list the prompts used for each task in the zero-shot setting. For the few-shot setting we follow a similar approach while adding 2-5 (depending on the task) examples at the beginning of each prompt.

### TEMPOWIC

Tweet 1: "In this bullpen, you should be able to ask why and understand why we do the things we do." @user #pitchstock2020 @user
Tweet 2: Castro needs to be the last bullpen guy to pitch.
Does the word "bullpen" mean the same thing in these two sentences?
Options: [ yes, no ]

### TWEETSIM

How similar are the following two tweets?
Tweet 1: India is With @republic #ImmortalSushant #FreeAnujNow #CantBlockRepublic #Nation_With_R_Bharat
Tweet 2: Trending On 5 Number Retweet And Comment For 1st #Nation_With_R_Bharat
Give the answer on a scale from 0 - 5, where 0 is "not similar at all" and 5 is "means the same thing".

### TWEETNERD

Based on the tweet is the definition of the target correct?
Tweet: No. 1 Eastern leads 3-0 at halftime against

Shawnee #njsoccer @user
Target: Shawnee
Definition: city in Pottawatomie County, Oklahoma, United States
Options: [ yes, no ]

### TWEETQA

Answer based on context:
Context: 5 years in 5 seconds. Darren Booth (@darbooth) January 25, 2013
Question: what site does the link take you to?
Answer:

### TWEETQG

Write a question based on this tweet and context.
Tweet: 5 years in 5 seconds. Darren Booth (@darbooth) January 25, 2013
Context: vine
Question:

### TWEETNER7

Entity Definition:
1.  corporation: Names of corporations (e.g. Google).
2.  creative_work: Names of creative works (e.g. Bohemian Rhapsody).
3.  event: Names of events (e.g. Christmas, Super Bowl).
4.  group: Names of groups (e.g. Nirvana, San Diego Padres).
5.  location: Names that are locations (e.g France).
6.  person: Names of people (e.g. Virginia Wade).
7.  product: Name of products (e.g. iPhone).

Identify and categorize named entities in the following tweet:
Tweet: New music coming soon via @Columbia_Records . . . . . . # columbia # newmusic # photooftheday # listentothis @ Columbia Records UK URL

### TWEETTOPIC

Which topics from the options below are present in the following tweet?
Tweet: Philadelphia clearly didn't take a page out of the @user Game 7 playbook of firing everything on net, make the opposing goalie beat you. There's 6 minutes left and the Flyers have 16 shots
Options:        [    arts_&_culture,    business_&_entrepreneurs, celebrity_&_pop_culture, diaries_&_daily_life,  family,  fashion_&_style,

| Model | Parameters | Link | Citation |
|---|---|---|---|
| RoBERTa_BASE | 123M | https://huggingface.co/roberta-base | Chung et al. (2022) |
| RoBERTa_LARGE | 354M | https://huggingface.co/roberta-large | |
| TimeLM_BASE | 123M | https://huggingface.co/cardiffnlp/twitter-roberta-base-2022-154m | Loureiro et al. (2022a) |
| TimeLM_LARGE | 354M | https://huggingface.co/cardiffnlp/twitter-roberta-large-2022-154m | |
| OPT_125M | 125M | https://huggingface.co/facebook/opt-125m | Zhang et al. (2022a) |
| OPT_350M | 350M | https://huggingface.co/facebook/opt-350m | |
| OPT-IML_1.3B | 1.3B | https://huggingface.co/facebook/opt-iml-1.3b | Iyer et al. (2023) |
| FlanT5_SMALL | 80M | https://huggingface.co/google/flan-t5-small | Chung et al. (2022) |
| FlanT5_BASE | 250M | https://huggingface.co/google/flan-t5-base | |
| FlanT5_XL | 3B | https://huggingface.co/google/flan-t5-xl | |
| FlanT5_XXL | 11B | https://huggingface.co/google/flan-t5-xxl | |
| text-ada-001 | 350M | https://platform.openai.com/docs/models/gpt-3 | Brown et al. (2020) |
| gpt-3.5-turbo | 175B | https://platform.openai.com/docs/models/gpt-3 | Brown et al. (2020) |

Table 9: Number of parameters, link to implementation, and reference for each model used in our experiments.

film_tv_&_video, fitness_&_health, food_&_dining, gaming, learning_&_educational, music, news_&_social_concern, other_hobbies, relationships, science_&_technology, sports, travel_&_adventure, youth_&_student_life ]

## TWEETSENTIMENT

What is the sentiment of this tweet regarding the target?
Tweet: #ArianaGrande Ari By Ariana Grande 80% Full {URL} #Singer #Actress URL
Target: #ArianaGrande
Options: [ 'strongly negative', 'negative', 'negative or neutral', 'positive', 'strongly positive']

## TWEETEMOTION

Which emotions from the options below are expressed in the following tweet?
Tweet: @user @AsYouNotWish Dont worry Indian army is on its ways to dispatch all Terrorists to Hell
Options: [ 'anger', 'anticipation', 'disgust', 'fear', 'joy', 'love', 'optimism', 'pessimism', 'sadness', 'surprise', 'trust' ]

## TWEETHATE

Classify the following tweet as hate speech based on the options.
Tweet: If you're angry and hate someone because of the color of their skin and you need to shoot someone then you're a p*nk*ss bitch. You're NOT SUPERIOR. YOU ARE A WEAK P*NK*SS B*TCH. F*CK YOU AND YOUR GUNS.
Options: [ hate_gender, hate_race, hate_sexuality, hate_religion, hate_origin, hate_disability, hate_age, not_hate ]

## TWEETINTIMACY

How intimate is the following tweet?
Tweet: thank u, nxt
Give the answer on a scale from 0 - 5, where 0 is "not intimate at all" and 5 is "very intimate".

## TWEETEMOJI100

Select the top 5 emojis that best describe the following tweet:
Tweet: Besties how tf do i get skinny in 12 days

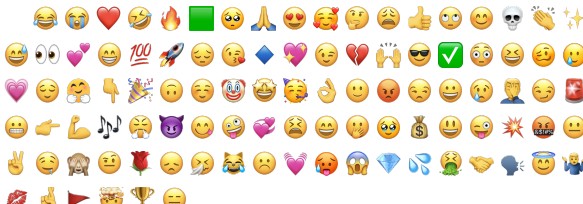