# OpenReview forum: "SuperTweetEval: A Challenging, Unified and Heterogeneous Benchmark for Social Media NLP Research"
_EMNLP/2023/Conference — EMNLP 2023 Findings_

### Official Review · Reviewer_B3Tx · 2023-08-04

**Soundness:** 3

**Excitement:**

4: Strong: This paper deepens the understanding of some phenomenon or lowers the barriers to an existing research direction.

**Paper Topic And Main Contributions:**

The paper introduces a benchmark consisting of multiple social media datasets (Twitter mainly) and tasks (10 classification and 2 generation) with the aim of providing a test bed for NLP models for social media. The paper also includes the results of the experiments of different models on the provided benchmark. The contributions of the paper are twofold: first, introducing a dataset “SuperTweetEval” which unifies 12 different datasets and tasks and second, a set of experiments to show how different models perform on the proposed tasks.

**Questions For The Authors:**

A. It is claimed that the paper is built upon TweetEval, however it seems to be excluding the datasets in that benchmark. Did you include all the datasets from TweetEval in your benchmark? If not what is your reason for not including them?

B. For Tweet-QG, what is the reason behind reframing the task and not using it as is?

C. For similarity and intimacy evaluation, what was the reason of not using Mean Absolute Error? Overall, I guess the benchmark could include more evaluation metrics for evaluating different aspects of the existing datasets.

D. For the Tweet-Emoji zero-shot or few-shot experiments, was it not possible to prompt the system to produce the top 5 related emojis? (I guess that should be doable at least for the GPT model)

E. You mentioned there were the issues that arose with the prompts tested, can you explain more?

F. Why are some missing numbers for some of the models for TweetQA and Tweet-QG in Table 3?

**Reasons To Accept:**

1. The paper builds upon the exiting benchmark (TweetEval) and extends it by modifying and creating new datasets to include the generation tasks as well.

2. The paper unifies 12 datasets (a mixture of classification and generation tasks) as a benchmark and conducts analysis on them and establish baselines and insights for further research in the domain of NLP for social media and tries to fill in the gap that currently exists in the field of social NLP.

3. The taxonomy of the task clusters presented in section 7 is interesting as it organizes the datasets into similar categories and that can be useful for research questions on specific types of tasks.

**Reasons To Reject:**

The main concern about the paper in my opinion is not using stronger existing models (the models are mainly the smaller size chosen from each family of models) on the tasks to better support the claim of the authors as the proposed benchmark can provide insights on how current models perform. Although the experiments conducted in the paper show the differences between different modeling strategies to some extent, it would be difficult to know how much room for improvement exists for different tasks and which aspects need to be worked on to further improve models. Therefore, it would be difficult to conclude that the benchmark is actually providing a challenging test bed for models.
If the best-performing LLM can do pretty well on these tasks, then there is not much room for improvements and therefore the benchmark cannot be seen as a strong and challenging test bed for future models. With the reported numbers one can not easily figure out if the benchmark poses challenging questions for researchers to answer.

Also, it would be really helpful to have an oracle (or even a proxy) for these tasks (either human oracle or any other relevant oracles) to provide the upper bound on the tasks as well. This can at least provide some intuition on how easy/difficult each task is.

**Reproducibility:**

4: Could mostly reproduce the results, but there may be some variation because of sample variance or minor variations in their interpretation of the protocol or method.

**Reviewer Confidence:**

4: Quite sure. I tried to check the important points carefully. It's unlikely, though conceivable, that I missed something that should affect my ratings.

---

> ### Author Rebuttal · Authors · 2023-08-29
>
> We thank the reviewer for the time and effort dedicated to review our manuscript. Below we identify the main points raised and attempt to address them.
>
> ***“The main concern about the paper in my opinion is not using stronger existing models (the models are mainly the smaller size chosen from each family of models) …” “... If the best-performing LLM can do pretty well on these tasks, then there is not much room for improvements and therefore the benchmark cannot be seen as a strong and challenging test bed for future models. …”***
>
> We added the baseline requested as reference baseline (ChatGPT), but please see our answer to reviewer 1 in relation to the justification of the models chosen, which is also included in Section 5.2 of the paper, and in particular why we refrained from evaluating closed models. And in particular to this concern, we would like to stress the utility of the benchmark irrespective of the performance of newer LLMs. In addition to its challenging nature, this dataset will serve as a test bed and for fine-tuning smaller models that may be developed in order to be used at a large scale, or simply as an analysis proxy. So far there are no social benchmarks such as the one we propose, which can serve to measure the progress of newer models and perhaps in the future, to prove that newer models can indeed be successfully used in all social media tasks (for which so far there is no evidence but our benchmark can help test these newer models).
>
> ***“Also, it would be really helpful to have an oracle (or even a proxy) for these tasks (either human oracle or any other relevant oracles) to provide the upper bound on the tasks as well. This can at least provide some intuition on how easy/difficult each task is.”***
>
> We agree that a human oracle could provide a useful estimate, although with some caveats as also acknowledged in the limitations section and explained in Tedeschi et al. (ACL 2023). Nevertheless, for most tasks this estimate can be computed from the inter-annotator agreement or similar metrics. We will add this information in the Appendix in order to provide a general idea on the tasks’ difficulty.
>
> ***“A. It is claimed that the paper is built upon TweetEval, however it seems to be excluding the datasets in that benchmark. Did you include all the datasets from TweetEval in your benchmark? If not what is your reason for not including them?”***
>
> With the exception of the SemEval-Emotion task, which utilises the same dataset but in a more realistic multi-label setting, none of the datasets present in TweetEval were used. The aim of SuperTweetEval is to expand on the original benchmark by creating new and challenging tasks while adjusting the difficulty of existing ones by better linking them to real-world tasks.  For example, the emoji prediction task of TweetEval consists of only 20 emojis while in SuperTweetEval we consider 100 emojis. Similarly, the sentiment analysis  task in SuperTweetEval is structured as topic based sentiment analysis in a 5 point scale, while TweetEval presents it as a classification approach. To achieve our goal we had to introduce new datasets that will be sufficiently diverse in terms of scope, while challenging for the current models available.
>
> ***“B. For Tweet-QG, what is the reason behind reframing the task and not using it as is?”***
>
> We are not sure that we fully understood the question. Tweet-QG is a modified version of the original Tweet-QA dataset structured as a question generation task, instead of the original question answering framing of Tweet-QA. We will make this clear in the paper to avoid confusion.
>
> ***“C. For similarity and intimacy evaluation, what was the reason of not using Mean Absolute Error? Overall, I guess the benchmark could include more evaluation metrics for evaluating different aspects of the existing datasets.”***
>
> For the similarity and intimacy evaluation we utilise the same metrics used in the original SemEval tasks (SemEval-2017 task 1 & Semeval 2023 task 9) . We agree that including more evaluation metrics is beneficial when studying an individual task. In the benchmark we only include a single metric for each task for simplicity, but we will include the option to compute alternative metrics for each task.
>
> ***“E. You mentioned there were the issues that arose with the prompts tested, can you explain more?”***
>
> The prompts used in our experiments are mostly modified versions of the prompts presented in the FLAN-T5 paper. However, for several tasks (e.g. Tweet-NER7, Tweet-Emoji) existing literature related to zero/few shot learning is limited (i.e., the performance may improve with a more optimised prompting methodology).
>
> ***“D. For the Tweet-Emoji zero-shot or few-shot experiments, was it not possible to prompt the system to produce the top 5 related emojis? (I guess that should be doable at least for the GPT model)”***
>
> We evaluated the selected models on the Tweet-Emoji task in zero/few -shot experiments, however their performance was very poor (not better than a random baseline).These prompts are available in Appendix C. Therefore, based on our prompts and models used, the results suggest that this task cannot be accomplished just by prompting, which may be due to several factors such as the large number of labels that may confuse the models, or the specificity of the task (the original T5 training corpus is not focused on social media and may not contain emoji). There may be more subtle ways to solve the task, but proposing new models was out of the scope of this paper, and we hope that our benchmark could encourage further research in this area.
>
> ***“F. Why are some missing numbers for some of the models for TweetQA and Tweet-QG in Table 3?”***
>
> For the Tweet-QA and Tweet-QG tasks we do not report results from the RoBERTa based models as only generative models were tested (i.e., only generative models were used on this task). We will add a footnote to explain this.

---

### Official Review · Reviewer_Anux · 2023-08-11

**Soundness:** 2

**Excitement:**

2: Mediocre: This paper makes marginal contributions (vs non-contemporaneous work), so I would rather not see it in the conference.

**Paper Topic And Main Contributions:**

The author proposes a challenging, unified and  heterogeneous benchmark for social media NLP research. As the authors, although the NLP techniques advance significantly, however, social media is still a challenging task. Authors test some models to prove their concept.

**Reasons To Accept:**

1. The authors bring an important task as the social media channel is important task for NLP area.
2. This paper provide a dataset which can be used to benchmark social media performance.

**Reasons To Reject:**

1. As the task is to test the ability of NLP models, there is no reason to skip large GPT-3 models, and I cannot give my judgement if I did not see the performance on large GPT-3. As we know, lots of challenging tasks now can be achieved with good performance.
2. As a benchmark dataset, I would like to see a much large corpus.

**Reproducibility:**

4: Could mostly reproduce the results, but there may be some variation because of sample variance or minor variations in their interpretation of the protocol or method.

**Reviewer Confidence:**

4: Quite sure. I tried to check the important points carefully. It's unlikely, though conceivable, that I missed something that should affect my ratings.

---

> ### Author Rebuttal · Authors · 2023-08-29
>
> We thank the reviewer for their review and the two issues raised.
>
> ***“As the task is to test the ability of NLP models, there is no reason to skip large GPT-3 models, and I cannot give my judgement if I did not see the performance on large GPT-3. As we know, lots of challenging tasks now can be achieved with good performance.”***
>
> We have added the results of the requested GPT baseline (please see the B. response to Reviewer 1). We can confirm that the tasks are challenging also for ChatGPT. Initially we refrained from using OpenAI’s larger/newer models due to: (1) the closed nature of proprietary models limits the research we can perform and it is questionable any insights may be actionable by the community; (2) the high probability that the GPT models may have already been trained on the datasets (including test splits) collected and the unfair advantage that this would give them; and (3) academic budget, i.e., we have access to a small number of medium-spec GPUs and some funding for using LLMs APIs, but properly running experiments with, e.g. GPT-4 would incur 10x the costs of our current experiment.
>
> ***“As a benchmark dataset, I would like to see a much large corpus.”***
>
> The SuperTweetEval benchmark consists of over 255,000 instances in total (the number of tweets being even larger), with the smallest dataset containing 1,000 entries. Therefore, despite the specialised nature of this dataset, SuperTweetEval is even comparable in size or larger than similar unified benchmarks such as  SuperGLUE - see Table 1 of the SuperGLUE paper for example for a comparison (Wang et al. 2020).

---

### Official Review · Reviewer_5755 · 2023-08-12

**Typos Grammar Style And Presentation Improvements:** n/a
**Soundness:** 4

**Excitement:**

4: Strong: This paper deepens the understanding of some phenomenon or lowers the barriers to an existing research direction.

**Missing References:**

n/a

**Paper Topic And Main Contributions:**

NLP for social media is much more immature than in general.  In part, this could be caused by a lack of competitive and public social media datasets.  To address this shortcoming the authors developed the SuperTweetEval dataset, which (like SuperGLUE vs GLUE) broadens the tasks and datasets available in TweetEval.  The new dataset collection includes existing datasets that are unchanged (7), datasets that have been adapted for the new benchmark (3), and entirely new datasets (2).  In all the dataset covers a total of 255,170 tweets across 12 task-specific datasets.  They also provide a number of baseline performance from multiple types of models including decoder only, encoder/decoder, and encoder only architectures.  The best fine tuned model is a BERT model that has been pretrained on twitter data, while the best zero/few-shot model was a FLAN-T5 model.  Limitations listed include a lack of non-English data, single dataset per task, and a limitation on the number of baselines provided.  In addition the work noted that in terms of ethical concers they attempted to follow all rules in respecting privacy of twitter users and in appropriately compensating any annotators.

**Questions For The Authors:**

A) how did you choose the additional tasks/datasets to include in SuperTwitterEval?  How did you choose those you retained from TwitterEval?  How do you know the new tasks are relevant to social media NLP researchers in general and are sufficiently difficult?

B) Why didn't you include stronger baselines to demonstrate the difficulty of SuperTwitterEval in the face of the latest LLMs?  For example chatgpt/gpt-4 instead of ada-001, or PaLM instead of Flan-T5.

**Reasons To Accept:**

The paper identifies a need in the publicly available social media datasets and is contributing a new dataset meant to fill that need.  This is timely for evaluating the gap in performance of the latest LLMs and related models. The overall dataset contributes a total of 5 new datasets if you consider the re-purposed datasets as new datasets.  The analysis includes a reasonable set of baselines, including a variety of model types and training types, but leaves plenty of room for contribution from future papers in terms of improving on the tasks.

**Reasons To Reject:**

The biggest concern I have is that it is hard to tell if the new dataset is really introducing a challenge to modern LLMs as claimed, because none of the latest models were evaluated on it.  The only OpenAI model evaluated was text-ada-001, which is a very old and the smallest size model available.  For example, evaluating at least ChatGPT on the tasks would have given a better idea on whether the new tasks are really going to be difficult for the latest models.  Even better would have been to see how GPT-4 does on the dataset.  Google's flan-T5-xxl is a good baseline, but what about Palm-Bison or something like that?

Another concern is that the paper never describes how the new tasks were selected.  In contrast, SuperGLUE describes a very practical way of determining their new set of tasks, "The remaining tasks were identified from those submitted to an open call for task proposals and were selected based on the difficulty for current NLP approaches." (SuperGLUE paper, see [1] below)  Such an open call or survey of those working in the field of social media NLP doesn't appear to have been done in this case (unless I missed this in the paper).  Are these new tasks really relevant to most social media NLP researchers or only to the authors?  How do they know?
It's also hard to know if the new datasets in SuperTweetEval were selected based on difficulty because we don't know what alternative datasets they considered or how they chose those that made it into the dataset.  Above concerns about using stronger baselines in the report compound the concerns, as we are not sure if these tasks were chosen due to actual difficulty or maybe just perceived difficulty by the authors.

[1] Quotes taken from Super Glue paper here: https://paperswithcode.com/dataset/superglue#:~:text=SuperGLUE%20is%20a%20benchmark%20dataset,language%20understanding%20technologies%20for%20English.

**Reproducibility:**

5: Could easily reproduce the results.

**Reviewer Confidence:**

4: Quite sure. I tried to check the important points carefully. It's unlikely, though conceivable, that I missed something that should affect my ratings.

---

> ### Author Rebuttal · Authors · 2023-08-29
>
> We thank the reviewer for the time and effort dedicated to review our manuscript. Below we identify the main points raised and attempt to address them.
>
> ***“A) how did you choose the additional tasks/datasets to include in SuperTwitterEval? How did you choose those you retained from TwitterEval? How do you know the new tasks are relevant to social media NLP researchers in general and are sufficiently difficult?”***
>
> Thank you for the details about the selection of SuperGLUE tasks. Indeed, our selection of tasks was somewhat different from how it was done in SuperGLUE. In particular, we aimed at having a diversity in terms of real-world tasks, which were carefully curated before being included in the benchmark. Then, once the task was decided, we gathered several datasets whenever available before deciding on the dataset to be introduced in the benchmark. Our goal was to provide challenging datasets, which were distinctive from the TweetEval benchmark focused on tweet multiclass classification. For example, for the hate speech task we went through a list of 13 different hate speech detection datasets on Twitter, finally deciding on one dataset based on the difficulty and coverage of the dataset, as well as the licence. In this case, the final dataset contains five distinct hate-speech classes, in contrast to other available options, e.g. two hate classes present in SemEval's 2019 task 5, and also utilises a more clear taxonomy than other available datasets (e.g. HateXplain, Mathew et al. AAAI, 2021).  A similar process was followed for all the tasks included.
>
> It is also worth noting that a few tasks present consist of more challenging/advanced versions of the original TweetEval benchmark (e.g. multi-label instead of multi-class emotion classification, or 100 emoji labels instead of 20 in emoji prediction).
>
> We will provide more details about the criteria we followed for selecting each individual dataset in the camera ready version, alongside size / usage comparisons with similar resources where available.
>
> ***“B) Why didn't you include stronger baselines to demonstrate the difficulty of SuperTwitterEval in the face of the latest LLMs? For example chatgpt/gpt-4 instead of ada-001, or PaLM instead of Flan-T5.”***
>
> We have computed ChatGPT results, available below. We initially refrained from using OpenAI’s larger/newer models due to: (1) the closed nature of these proprietary models limits the research we can perform and there is high probability that the GPT models may have already been trained on the datasets (including test splits) collected and the unfair advantage that this would give them; and (2) budget, i.e., we have access to a small number of medium-spec GPUs and some funding for using LLMs APIs, but properly running experiments with, e.g. GPT-4 would incur 10x the costs of our current experiments.
>
> What follows are the zero-shot results of ChatGPT compared to the results reported in the paper:
> |                 | ChatGPT | Best zero-shot | Best fine-tuned |
> | --------------- | ------- | -------------- | --------------- |
> | TempoWic        | 0.6495  | 0.6214         | **0.6841**          |
> | SemEval-Emotion | 0.4517  | 0.3277         | **0.5853**          |
> | Tweet-Hate      | 0.6342  | 0.5105         | **0.8254**          |
> | Tweet-Intimacy  | 0.4182  | 0.2822         | **0.6895**          |
> | TweetQA         | 0.3290  | 0.5388         | **0.6609**          |
> | Tweet-NERD      | 0.6997  | 0.7556         | **0.853**           |
> | TweetQG         | 0.2325  | 0.2176         | **0.4404**          |
> | Tweet-Sentiment | 0.4299  | 0.283          | **0.5464**          |
> | TweetSim        | 0.6894  | 0.5946         | **0.7464**          |
> | TweetTopic      | 0.5477  | 0.3625         | **0.5884**          |
> | TweetEmoji      | 0.1650  | \-             | **0.3564**          |
> | Tweet-NER7      | 0.1804  | \-             |  **0.604**           |
>
> As can be observed, ChatGPT performs better than the other models tested in a similar zero-shot setting, with TweetQA and Tweet-NERD being the only tasks where it fails to outperform FLAN-T5-XXL (scores: 0.5388 and 0.7556 respectively). However, when compared to the best performing fine-tuned models, ChatGPT fails to achieve greater scores to the fine-tuned models with differences ranging between 0.03 (TempoWic) to 0.4236 (Tweet-NER7) in favour of fine-tuning.
>
> The results reinforce our conclusion that there is room for improvement and that SuperTweetEval remains challenging also for recent closed-source LLMs.
>
> Regarding open-source models, FLAN-T5 was selected as it has been shown to be the strongest instruction-fine tuned available for this type of tasks at the time of the writing (while we acknowledge that it is hard to compare across models and reviewers may believe there are stronger models). Thank you for the suggestion of PaLM, we will also add these results in the final version of the paper.

---

### Meta-Review · Area_Chair_hYFg · 2023-09-18

**Recommendation:** 4

**Metareview:**

The work presents a benchmark for NLP evaluation in social media, SuperTweetEval.

The reviewers scores vary considerable from borderline/mediocre to good/strong with respect to both soundness and excitement. One major point of agreement among reviewers has to do with the lack of evaluation of latest/best-performing models to date to provide a wider picture of the challenges & limites presented by the benchmark. The authors add models such as Chat-GPT to the papter as requested in order to provide a wider picture. The authors also provide further justifications during rebuttal for the models selected. This aspect should be reinforced in the camera-ready version of the paper. The authors should clearly describe how the datasets in SuperTweetEval were selected, how are they relevant to social media NLP research, and how they compare to previously existing similar resources.


The points of divergence are: the size of the dataset and the lack of human baselines. The authors address this issues during rebuttal and in my opinion should not preclude publication.

---

### Decision · Program_Chairs · 2023-10-07

**Decision:**

Accept-Findings

**Comment:**

The work presents a benchmark for NLP evaluation in social media, SuperTweetEval.

The reviewers scores vary considerable from borderline/mediocre to good/strong with respect to both soundness and excitement. One major point of agreement among reviewers has to do with the lack of evaluation of latest/best-performing models to date to provide a wider picture of the challenges & limites presented by the benchmark. The authors add models such as Chat-GPT to the papter as requested in order to provide a wider picture. The authors also provide further justifications during rebuttal for the models selected. This aspect should be reinforced in the camera-ready version of the paper. The authors should clearly describe how the datasets in SuperTweetEval were selected, how are they relevant to social media NLP research, and how they compare to previously existing similar resources.


The points of divergence are: the size of the dataset and the lack of human baselines. The authors address this issues during rebuttal and in my opinion should not preclude publication.